# Tuberculosis susceptibility in genetically diverse mice reveals functional diversity of neutrophils

Marietta M Ravesloot-Chavez[1], Erik Van Dis[2], Douglas Fox[2], Andrea Anaya-Sanchez[1], Scott Espich[3], Xammy Huu Nguyenla[2], Sagar Rawal[2], Helia Samani[2], Mallory Ballinger[4], Henry F Thomas[4], Dmitri I Kotov[2], Russell E Vance[2], Michael W Nachman[4], Sarah A Stanley[2,3]*

[1]Department of Plant and Microbial Biology, University of California, Berkeley, Berkeley, United States; [2]Division of Immunology and Pathogenesis, Department of Molecular and Cell Biology, University of California, Berkeley, Berkeley, United States; [3]School of Public Health, Division of Infectious Disease and Vaccinology, University of California, Berkeley, Berkeley, United States; [4]Department of Integrative Biology, University of California, Berkeley, Berkeley, United States

*For correspondence:
sastanley@berkeley.edu

## eLife Assessment

This study provides **valuable** insights into the host's variable susceptibility to *Mycobacterium tuberculosis*, using a novel collection of wild-derived inbred mouse lines from diverse geographic locations, along with immunological and single-cell transcriptomic analyses. While the data are **convincing**, a deeper mechanistic investigation into neutrophil subset functions would have further enhanced the study. This work will interest microbiologists and immunologists in the tuberculosis field.

**Abstract** Tuberculosis is a heterogeneous disease in humans with individuals exhibiting a wide range of susceptibility. This heterogeneity is not captured by standard laboratory mouse lines. We used a new collection of 19 wild-derived inbred mouse lines collected from diverse geographic sites to identify novel phenotypes during *Mycobacterium tuberculosis* (*Mtb*) infection. Wild-derived mice have heterogeneous immune responses to infection that result in differential ability to control disease at early time points. Correlation analysis with multiple parameters including sex, weight, and cellular immune responses in the lungs revealed that enhanced control of infection is associated with increased numbers of CD4 T cells, CD8 T cells, and B cells. Surprisingly, we did not observe strong correlations between IFN-γ production and control of infection. Although in most lines high neutrophils were associated with susceptibility, we identified a mouse line that harbors high neutrophil numbers yet controls infection. Using single-cell RNA sequencing, we identified a novel neutrophil signature associated with failure to control infection.

## Introduction

*Mycobacterium tuberculosis* (*Mtb*) infection causes a wide range of disease outcomes in humans. Whereas the majority of infected individuals never develop active disease and are able to control infection for their lifetimes, some individuals are highly susceptible to infection and develop active tuberculosis (TB). Patients with active disease show heterogeneity in disease progression. Additionally, the

timing of disease onset after initial infection and disease outcome varies from individual to individual (*Cadena et al., 2017*). This heterogeneity in human responses suggests that TB is a complex genetic disease. Surprisingly, there is only a limited understanding of the genetic factors that contribute to disease progression or control of infection. Indeed, GWAS studies have identified a small number of genetic factors that underlie the variability of responses to *Mtb* in humans including interferon-γ (*IFNG*), interferon-γ receptor (*IFNGR1*), interleukin-12 subunit p40 (*IL12B*), and interleukin-12 receptor b-1 chain (*IL12RB1*) (*Qu et al., 2011*). However, many questions remain regarding what factors determine susceptibility to infection. Despite the clear importance of IFN-γ for control of mycobacterial infection, new vaccine candidates that elicit IFN-γ producing T cells in human volunteers do not elicit complete protection from infection, suggesting that additional protective mechanisms that have yet to be identified may be essential (*Tameris et al., 2013*; *Tait et al., 2019*). Similarly, several immune signatures have been associated with active disease in TB patients, including type I IFN and neutrophils (*Donovan et al., 2017*). However, there has been little consensus on signatures that will predict the course of disease in patients, possibly in part due to the heterogeneity of mechanisms that lead to active disease (*Nogueira et al., 2022*).

The murine model of TB infection is a useful tool for analysis of host immune responses to infection due to the abundance of available reagents, ability to precisely control experimental conditions, and ease of genetic manipulation. Importantly, the mouse model captures much of what is known about human immune responses to *Mtb*, including the importance of CD4 T cells and IFN-γ for immune control of infection (*Green et al., 2013*; *Caruso et al., 1999*; *Flynn et al., 1993*; *Scanga et al., 2000*; *Cooper et al., 1993*). However, current mouse models fail to recapitulate several aspects of *Mtb* infection, including spontaneous resolution of infection as well as the widely varying susceptibility and patterns of disease seen in humans. Although there are minor differences in susceptibility to infection, most commonly used mouse lines all exhibit similar infection dynamics and characteristics. However, despite the limited diversity of common laboratory lines, comparison of divergent phenotypes followed by genetic mapping has identified loci that contribute to susceptibility to mycobacterial infection, leading to valuable insights into immune factors that influence susceptibility to infection. An example is the characterization of the *Sst1* locus, which regulates type I IFN signaling through the gene *Sp140*, a repressor of type I IFN (*Ji et al., 2019*; *Pan et al., 2005*; *Kramnik et al., 2000*; *Kramnik et al., 1998*). Excessive production of type I IFN in *Sst1* susceptible (*Sst1$^s$*) mice, which have a loss of function mutation of *Sp140*, elicits increased IL-1 receptor antagonist (IL1-Ra) expression which competes with IL-1 for binding to the IL-1R. IL-1 is crucial for control of infection and the functional deficiency in IL-1 signaling in *Sst1$^s$* mice results in increased susceptibility to *Mtb* (*Ji et al., 2019*; *Ji et al., 2021*). Analysis of the phenotype of these highly susceptible mice led to the discovery of a mechanism by which type I IFN is detrimental to host infection. These studies suggest that increasing the genetic diversity in mouse models will lead to the discovery of additional genetic loci that are important for modulating the outcome of *Mtb* infection.

Historically, most *Mtb* infectious studies in the mouse model have been conducted using only a handful of lines: C57BL/6 (B6), BALB/C, and C3Heb/FeJ. In order to increase genetic diversity, new collections of mice have been established. The Collaborative Cross (CC) model was founded by crossing five common laboratory lines and three wild-derived lines (*Churchill et al., 2004*). Further breeding of the CC lines created the Diversity Outbred (DO) mouse collection, which is maintained as outbred population through random crossings. This is currently the most diverse mouse resource available (*Svenson et al., 2012*). Both cohorts have been used for the study of *Mtb* infection and display a range of disease phenotypes, including enhanced control or susceptibility to infection compared to standard laboratory lines (*Smith et al., 2016*; *Niazi et al., 2015*; *Gopal et al., 2013*). Infection of the DO cohort with *Mtb* revealed correlations between disease pathology and neutrophil influx, recapitulating features seen in human lung infections, in addition to identifying new inflammatory biomarkers that correlate with active human disease (*Niazi et al., 2015*; *Koyuncu et al., 2021*). Furthermore, these cohorts have been used to identify the genetic basis of disease responses. For example, the susceptibility of the CC042 line is associated with the loss of *Itgal* expression, an integrin required for T cell trafficking to the lung, resulting in decreased T cell trafficking and IFN-γ production in the lungs of infected animals (*Smith et al., 2019*).

Studies using CC and DO mice demonstrate the power of genetically diverse mice for identifying novel host regulators of immunity to *Mtb* (*Smith et al., 2022*). However, it is unclear whether these

collections capture the full spectrum of genetic diversity that exists in house mouse populations. To increase the genetic diversity of house mice available for the analysis of complex phenotypes, researchers recently created a set of 19 new inbred lines derived from wild mice collected across different geographic regions (*Phifer-Rixey et al., 2018*; *Moeller et al., 2018*; *Suzuki et al., 2020*). Unlike CC and DO mice, these new lines are all from the *Mus musculus domesticus* subspecies of house mouse. The use of a single subspecies to establish this cohort is beneficial, as it limits allelic conflicts that can arise from crossing different subspecies. This allows for the maintenance of natural genetic heterogeneity and makes the presence of genomic 'blind spots' – regions of the genome that are the same in all the lines – less likely (*Salcedo et al., 2007*; *Yang et al., 2011*). The mice that constitute this collection were originally collected across a latitudinal gradient from Brazil to Canada and show significant phenotypic diversity, including size, body mass index, and lipid and glucose metabolism (*Phifer-Rixey et al., 2018Phifer-Rixey et al., 2018*). Each line has been inbred through sib–sib mating for at least 10 generations, providing the benefit of the genetic reproducibility of laboratory-derived lines.

This new collection of wild-derived inbred lines represents a unique resource for identifying genetically diverse mice with enhanced or diminished susceptibility to *Mtb* infection and eventually, genes that control the host response to *Mtb* infection. Here we characterize 17 lines from this collection to identify lines that exhibit diminished control of *Mtb* infection at an early time point after infection (21 days) and analyze the accompanying immune responses. This is the first report of the use of this collection of mice for infectious disease or immunology-related phenotypes. Infection of the genetically diverse mouse lines reveals a range of responses with several lines showing higher susceptibility to infection compared with standard B6 mice. New susceptible lines of mice will be useful for identifying new mechanisms of susceptibility. In addition, we have identified heterogeneity in immune responses to infection that are distinct from known mouse lines, including robust control of infection in the absence of a strong Th1 T cell response and effective control of infection in the presence of elevated neutrophils. Our findings suggest that these mice will be useful not only for investigation of additional phenotypes in the context of TB infection but also may be useful for identifying novel immunological phenotypes more generally.

## Results

### Wild-derived mouse lines display variability in immune responses and susceptibility to *M. tuberculosis* infection

The cohort of wild-derived mice used in this study was established using outbred mice originally collected from five different localities across the Americas: Edmonton (EDM), Alberta, Canada; Saratoga Springs (SAR), New York, United States of America; Gainesville (GAI), Florida, United States of America; Tucson (TUC), Arizona, United States of America; and Manaus (MAN), Amazonas, Brazil (*Figure 1A*). Multiple individuals were trapped at each locale and used to create multiple, independent lines from an individual location. These mice exhibit a variety of phenotypic differences, including body weight, body length, body mass index (BMI), and glucose and lipid metabolism (*Phifer-Rixey et al., 2018*). Although there is within-group heterogeneity, separate lines of mice collected from a single location are in general more closely related genetically than mice collected from different locales (*Ferris et al., 2021*). We hypothesized that the genetic diversity of these mouse lines would contribute to heterogeneity in their capacity to control *Mtb* infection. We focused on a single time point, 21 days post-infection, to specifically identify early loss of control. Observing enhanced control of infection is also possible, though less likely to be observed at this relatively early time point after the onset of adaptive immunity. Mice were challenged with aerosolized *Mtb*, and the bacterial burden was assessed by colony-forming units (CFU) after 3 weeks of infection. CFU was normalized to inoculation dose day 1 post-infection to correct for any variability in inhaled bacterial load between mouse lines with different activity levels, sex, or different days of infection. The median bacterial burden varied over 100-fold between mice from different locales (*Figure 1B*). Surprisingly, individual mouse lines that originated from the same locale demonstrated a wide variation in susceptibility, suggesting that genetic heterogeneity within the closely related lines can influence the outcome of infection. Bacterial burdens in the lungs of mice from the MANC line are significantly higher compared to the closely related MANA, MANB, and MANE lines which are approximately as resistant as the standard

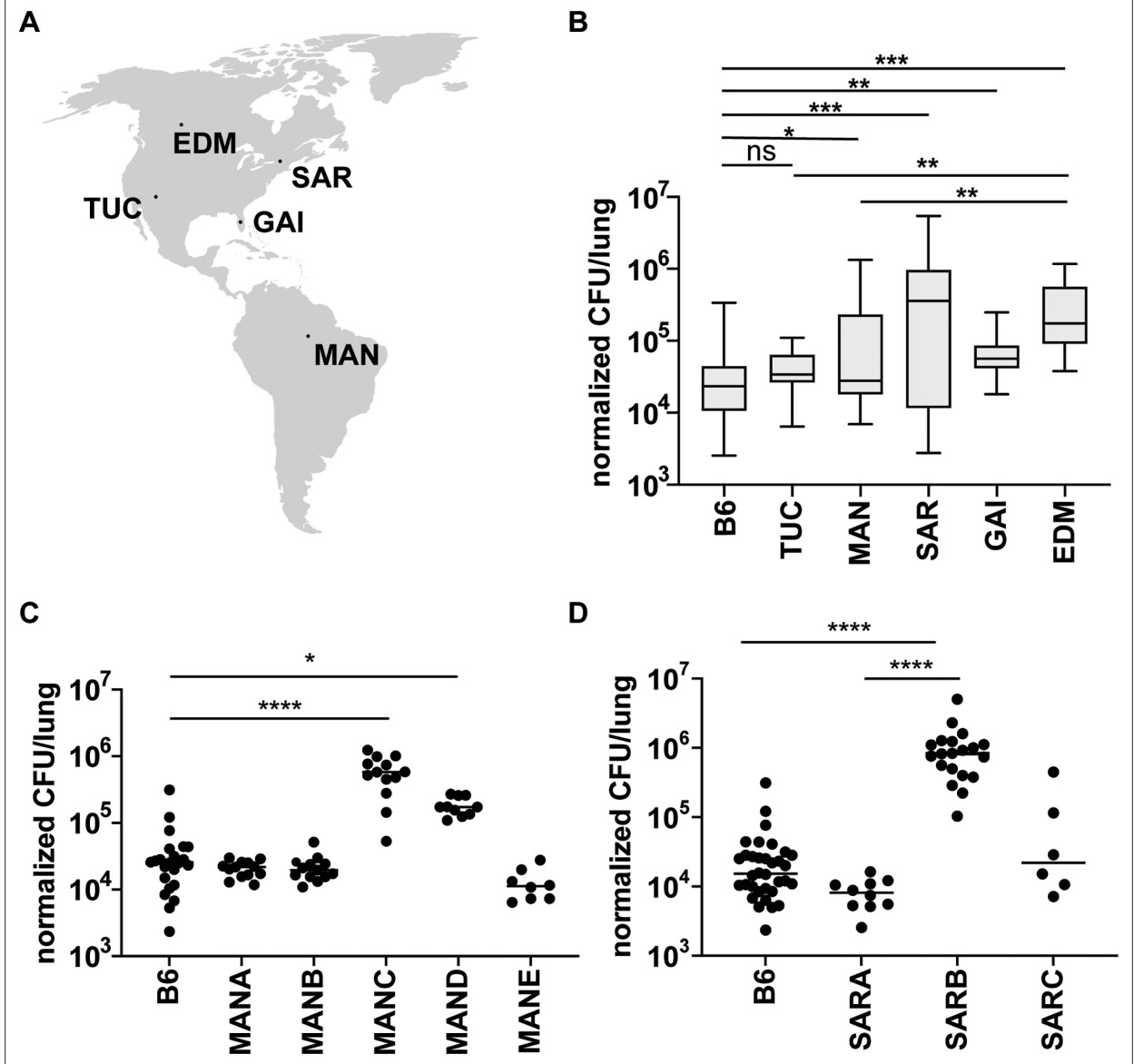

**Figure 1.** Mouse lines from different geographical regions show variable susceptibility to *Mtb* infection. (**A**) Mice were collected from five different sites across the Americas and inbred to generate individual genetic lines from each location. Map obtained from https://simplemaps.com/resources/svg-world. (**B**) Bacterial burden in the lungs of the mice derived from the different locations at 21 days post-infection with aerosolized *Mtb*. Comparison of the susceptibility among genetically distinct Manaus lines (**C**) or Saratoga lines (**D**) infected with aerosolized *Mtb*. The average inoculum dose is 400 CFU/mouse. CFU data was normalized to CFU enumeration on day 1 to correct for variation in inoculum dose between experimental days. EDM: Edmonton, SAR: Saratoga Springs, TUC: Tucson, GAI: Gainesville, MAN: Manaus. Data represent 2–6 independent experiments for each mouse line with 6–34 mice per line. Each dot represents a mouse. The p values were determined using a Kruskal–Wallis ANOVA. *p < 0.05, **p < 0.01, ***p < 0.001, ****p < 0.0001.

The online version of this article includes the following figure supplement(s) for figure 1:

**Figure supplement 1.** Variation in susceptibility to infection in wild mice does not depend on mouse weight or sex.

B6 (*Figure 1C*). Mice from the MAND line show an intermediate phenotype. We also observed significant variability in bacterial burdens in mice from Saratoga Springs. Whereas SARA and SARC mice are at least as resistant to infection as the standard B6, SARB lines harbor significantly more bacteria in the lungs at 3 weeks post-infection (*Figure 1D*). Smaller differences in susceptibility are found in the lines from Tucson, Gainesville, and Edmonton (*Figure 1—figure supplement 1A–C*).

A low BMI has been shown to be a risk factor for TB in humans (*Lönnroth et al., 2010*). Because the different lines of wild mice vary considerably in body weight even when provided the same diet,

we examined whether differences in body weight could account for differences in the ability to control infection. For this analysis, we focused on the Saratoga Springs and Manaus lines of mice where we observed the highest amount of variability in the control of infection. We found that there were significant differences in the starting weight of mice prior to infection when comparing the different lines, with a trend toward increased body weight in mice from Saratoga Springs (*Figure 1—figure supplement 1D*). However, we did not find a correlation between body weight and TB control in these well-nourished mice (*Figure 1—figure supplement 1E*). Similarly, previous reports have suggested that sex plays a role in susceptibility to infection, with females being more susceptible to TB infection in both mice and humans. We also observed no differences in bacterial burden at 21 days post-infection between male and female mice (*Figure 1—figure supplement 1F*). This is not surprising, as demonstrated differences in the ability of male and female mice to control infection emerged only at late time points ~200 days after infection (*Dibbern et al., 2017*).

We next reasoned that genetically determined differences in immune responses to infection might underlie the differences in susceptibility observed between different lines of wild-derived mice. To identify immune correlates of bacterial burden in the lungs of the infected wild mouse lines, we evaluated the presence of major immune cells thought to play a role in infection using flow cytometry in parallel with CFU counts at 21 days post-infection. We observed variability in the relative proportions of macrophages, neutrophils, CD4 T cells, CD8 T cells, and B cells when comparing individual lines (*Figure 2A–E*, *Figure 2—figure supplement 1*). Because the strains are very different in body size, in this case, proportion is more informative than absolute cell numbers. The most striking differences observed were in CD11b+Ly6G+ neutrophils, both when comparing individual lines isolated from a single geographic site and when comparing mice originally derived from different locales (*Figure 2A–E*).

In order to determine whether observed differences in the proportions of immune cells may drive the outcome of infection, we examined correlations between cell type and bacterial burden. We found significant correlations between the proportion of CD3+, CD4+, CD8+, and B220+ cells with control of infection, with larger proportions of these cells corresponding to lower CFU in the lung (*Figure 2F–I*, *Figure 2—figure supplement 2A*). In contrast, we found that the proportion of neutrophils in the lung could explain approximately 60% of the variation seen in the wild-derived mice, with higher neutrophil burden associated with higher CFU in the lungs (*Figure 2J*). This is consistent with numerous studies of susceptible mice on the B6 background that have pointed to a role for excessive infiltration of neutrophils in promoting lung pathology and susceptibility to *Mtb* infection (*Kimmey et al., 2015*; *Mishra et al., 2017*; *Nair et al., 2018*; *Dorhoi et al., 2010*). Production of IFN-γ by T cells is known to be important for control of infection (*Green et al., 2013*; *Cadena et al., 2016*; *Mogues et al., 2001*; *Bold and Ernst, 2012*). As expected, B6 mice exhibit a high proportion of IFN-γ producing CD4 T cells and lower bacterial burden in the lungs compared to several wild mice lines (*Figure 2K*). This was particularly evident in mice from Saratoga Springs (SAR). The SARA line controls bacterial CFU in the lungs in the presence of very few IFN-γ producing CD4 T cells (*Figure 2—figure supplement 2B*). No correlations were found between bacterial burden and the proportion of IFN-γ producing CD8 T cells or CD11b+ cells (*Figure 2L, M*). The overall variation in bacterial burden and immune cells present in the lung among and between the locales indicates the heterogeneity in underlying immune responses to *Mtb* infection. The notable differences between the wild-derived mouse lines and B6 mice show the potential for identifying novel mechanisms of susceptibility to infection in these mice.

## Macrophages from Manaus lines show no intrinsic defects that correlate with control of infection

Despite being closely genetically related, the Manaus lines show distinct susceptibilities *to Mtb*. Whereas most of the tested lines control infection in the lungs as well as B6, the MANC line is significantly more susceptible to infection (*Figure 1C*). We therefore focused on these lines for a more in-depth immunological analysis. First, we tested whether impaired bacterial restriction in the MANC line can be explained by differences in cell intrinsic responses in host cells. Macrophages are central to *Mtb* infection, serving both as the host cell, the cell responsible for killing the bacteria, and an important source of cytokines that regulate infection (*Cohen et al., 2018*; *Srivastava et al., 2014*; *Domingo-Gonzalez et al., 2016*; *Ravesloot-Chávez et al., 2021*). To elucidate whether macrophage

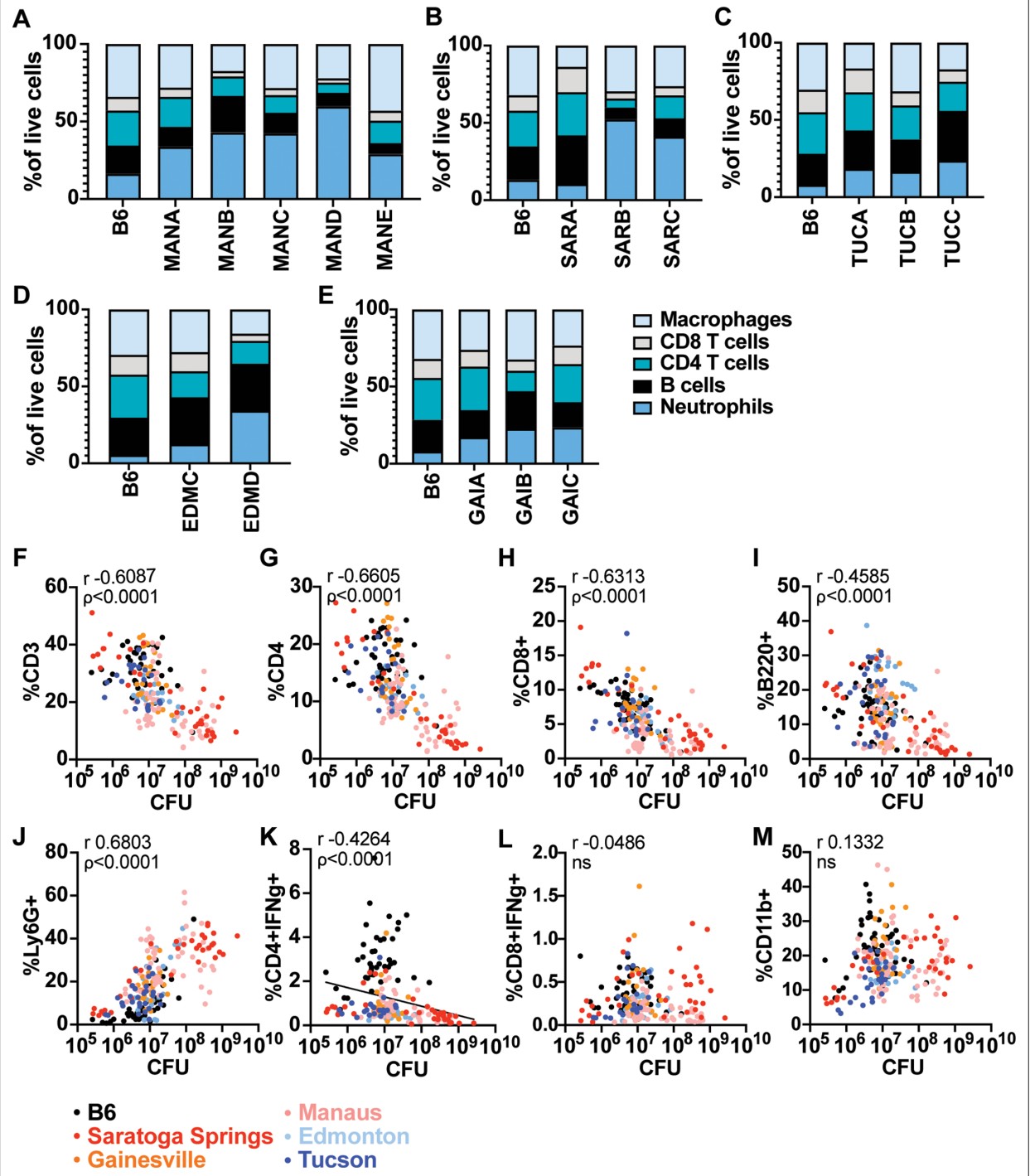

**Figure 2.** Genetic diversity translates into variability in bacterial burden and immune cells in the lung during *Mtb* infection. Cellular percentages were enumerated in the lungs using flow cytometry at 21 days post-infection with *Mtb* between and within mouse lines from different locales (**A–E**). The correlation between immune cell type (as shown in **A–E**) and CFU in the lung is depicted for individual mice (**F–M**). Data represent 2–6 independent experiments for each mouse line with 6–34 mice per line. The p value was determined using two-tailed Spearman correlations.

The online version of this article includes the following figure supplement(s) for figure 2:

**Figure supplement 1.** Cell percentages from flow cytometry-based analysis in the lungs of infected wild mice relative to B6.

**Figure supplement 2.** The proportion of IFN-γ producing CD4 T cells in the lung does not correlate with bacterial burden.

**Figure supplement 3.** Immune cell composition in the lungs of infected mice.

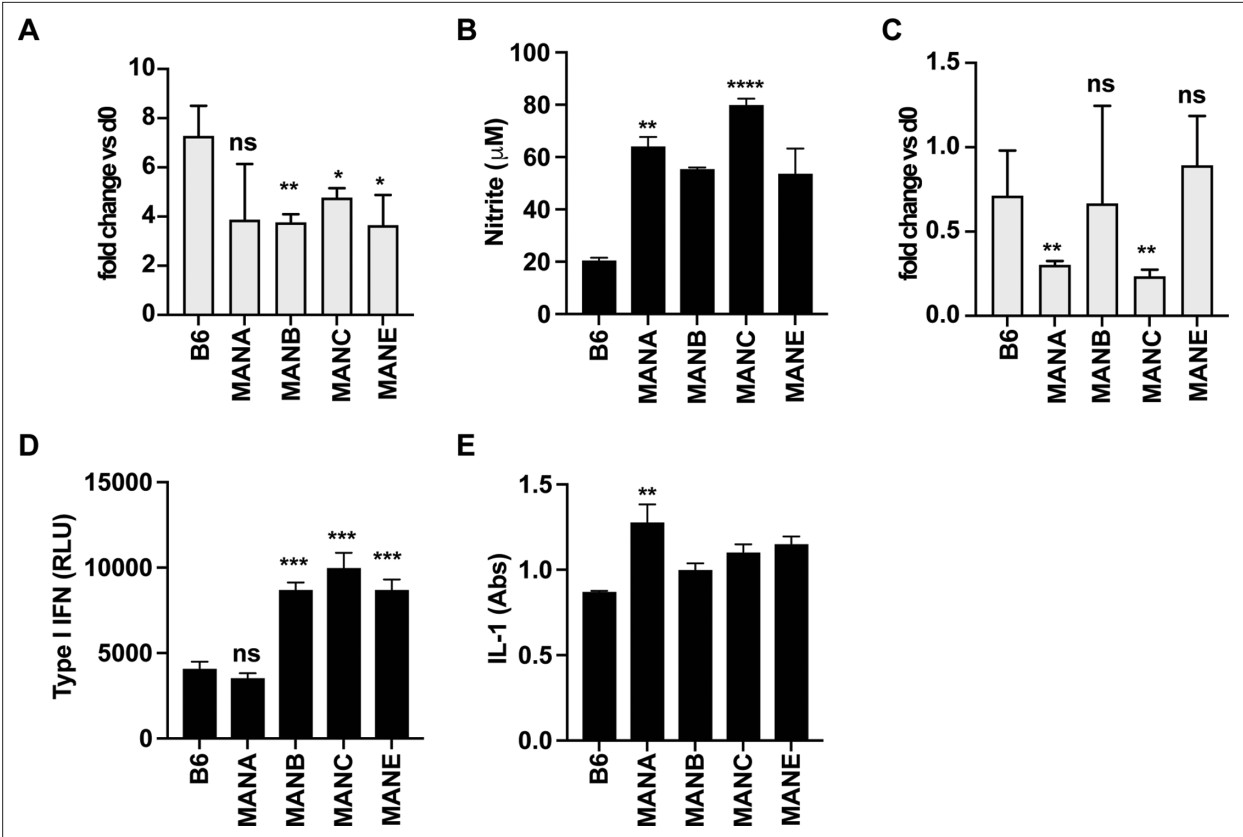

**Figure 3.** Intrinsic control of infection and response to IFN-γ is not impaired in macrophages from wild-derived mice. (**A**) Bone marrow-derived macrophages (BMDM) isolated from indicated mouse lines were infected with *Mtb* and bacterial growth was determined at 4 days post-infection by enumerating CFU. (**B**) The presence of nitrite in the supernatant was determined by Griess assay at 48 hr post-infection. (**C**) IFN-γ-activated BMDM were infected with *Mtb* and CFU were measured on day 4 post-infection. Production of type I IFN (**D**) or IL-1 (**E**) was measured at 24 hr post-infection with reporter cell lines. Data are representative of two independent experiments. The p values were determined using a Kruskal–Wallis ANOVA. *p < 0.05, **p < 0.01, ***p < 0.001 ****p < 0.0001.

responses differ between mouse lines, bone marrow-derived macrophages (BMDM) from four Manaus lines were prepared and infected with *Mtb* followed by quantification of bacterial burden and cytokine production. Macrophages from all the tested lines, including B6, allow the growth of *Mtb* in the absence of exogenous cytokine stimulation (*Figure 3A*), with none of the lines exhibiting a loss of control relative to B6. Indeed, we observed a slight decrease in bacterial replication in MANB, MANC, and MANE relative to B6 (*Figure 3A*). Additionally, all macrophages responded similarly to IFN-γ stimulation, producing significantly more nitric oxide (NO) upon infection than is observed with B6 (*Figure 3B*). Similarly, we did not observe any defects in IFN-γ-based control of infection at the macrophage level as all MAN lines controlled infection as well or better than the standard B6 (*Figure 3C*). Thus, differences in cell intrinsic control of infection that are evident ex vivo do not obviously play a role in the susceptibility of the MANC line to *Mtb* infection.

Type I IFN is known to be a major driver of susceptibility in mice, and possibly humans (*Ji et al., 2019*; *Dorhoi et al., 2014*; *Desvignes et al., 2012*; *Antonelli et al., 2010*; *Berry et al., 2010*; *Zak et al., 2016*; *Singhania et al., 2018*; *Scriba et al., 2017*; *Llibre et al., 2019*). Macrophages from *Sp140*⁻/⁻ mice that are highly susceptible to infection produce excessive type I IFN upon immune stimulation, recapitulating the high type I IFN/low IL-1 signaling phenotype of these animals in vivo. We next tested whether increased type I IFN or impaired IL-1 production by infected macrophages could underlie the high bacterial burden. Macrophages were infected with *Mtb* and levels of type I IFN were assessed at 24 hr post-infecting using interferon-stimulated response element (ISRE)-L929 reporter cells (*Jiang et al., 2005*). We found that macrophages isolated from three MAN lines (MANC, MANB, and MANE) produced significantly more type I IFN than B6 or MANA (*Figure 3D*); however, these

findings did not correlate with the ability to control infection in vivo (*Figure 1C*). We also measured IL-1 bioactivity in the supernatants of infected macrophages using the HEK-Blue IL-1R cell line and found that in vitro production of IL-1 was comparable to other Manaus genotypes and to B6. Taken together, unlike the majority of susceptible mouse lines, none of the Manaus lines appear to have major macrophage intrinsic defects observable that would explain the observed differences in control of infection in vivo.

### Heterogeneity in neutrophil phenotypes drives differences in bacterial burden

Neutrophils possess potent antimicrobial mechanisms, yet their role in host defense during *Mtb* infection is not well defined. In human TB, neutrophils are generally associated with active disease and pathogenesis (*Gopal et al., 2013*; *Eum et al., 2010*; *Lowe et al., 2012*). The vast majority of mouse lines with excessive accumulation of neutrophils in the lung have high bacterial burdens in the lung and are susceptible to infection (*Niazi et al., 2015*; *Mishra et al., 2017*; *Nandi and Behar, 2011*; *Eruslanov et al., 2005*; *Lyadova et al., 2010*; *Marzo et al., 2014*; *Keller et al., 2006*). Interestingly, the majority of the Manaus lines show high neutrophilic influx to the lungs compared to B6 mice, despite the strong variability in bacterial burden (*Figures 1C and 2A*). In general, the proportion of neutrophils in the lung is a clear predictor of susceptibility to infection across all lines examined (*Figure 2J*, *r* = 0.6803). Indeed, we observed strong correlations in the proportion of neutrophils in the lungs with susceptibility to infection in B6 mice, which overall have robust control of infection (*Figure 2—figure supplement 3A*) as well as mice from Saratoga Springs, which encompass both susceptible and resistant lines (*Figure 2—figure supplement 3B*). However, when examining mice from Manaus, we found a significantly decreased correlation in the proportion of neutrophils in the lungs and the total CFU, in line with our initial observation that these mice tend to have a higher proportion of neutrophils in the lungs than B6 despite variability in CFU (*Figure 2—figure supplement 3C*). These data suggest that mice from Manaus may have phenotypic heterogeneity in neutrophil populations that results in diminished correlation between neutrophils and susceptibility to infection.

   Recently, studies have suggested that neutrophils are functionally heterogeneous cells and can participate in the inflammatory response in both protective and deleterious roles. This prompted us to investigate the phenotypes of neutrophils in the Manaus lines during infection. We first sought to recapitulate our CFU and neutrophil data in experiments where we focused only on B6 and Manaus lines. Mice from Manaus controlled infection with *Mtb* as well as B6 mice, as measured by CFU in the lungs at 25 days post-infection, with the exception of the MANC line (*Figure 4A*). All Manaus mice have a higher proportion of neutrophils in the lungs (*Figure 4B*). In addition, when comparing the absolute number of neutrophils in the lungs of infected mice, MANB and MANC had elevated, roughly equivalent, absolute numbers of neutrophils in the lungs (*Figure 4C*), despite very different bacterial burdens (*Figure 4A*). We next tested whether the high neutrophilic influx in the lungs of MANC mice drives susceptibility to infection. We treated mice with an antibody targeting Ly6G to deplete neutrophils. Antibody depletion resulted in a dramatic drop of ~40-fold in CFU in the lungs of MANC mice (*Figure 4D, E*). In contrast, depleting neutrophils from B6 or MANB mice had a more modest effect on CFU burden of two- to threefold (*Figure 4F*). Thus, in MANC mice, neutrophils are an important driver of susceptibility.

### Granuloma-like structures in MANC mice are dominated by neutrophils and have a paucity of both macrophages and T cells

Because both MANB and MANC mice have elevated levels of neutrophils in the lungs relative to B6, we next sought to determine whether the spatial localization of neutrophils differed between these lines. To do so, we used immunofluorescence microscopy to characterize granuloma-like structures in MANB and MANC mice infected with *Mtb.* Both MANB and MANC lesions contained large numbers of neutrophils as determined by staining with both CD11b and Ly6G antibody. Whereas neutrophils appeared as discrete cells within lesions of MANB mice (*Figure 4G*), neutrophils in MANC lesions appeared to form large aggregates of cells (*Figure 4H*). In addition, there was twice the amount of neutrophil staining per area in MANC lesions relative to MANB (*Figure 4I*) and far fewer CD4 T cells within MANC lesions than within MANB lesions (*Figure 4J*).

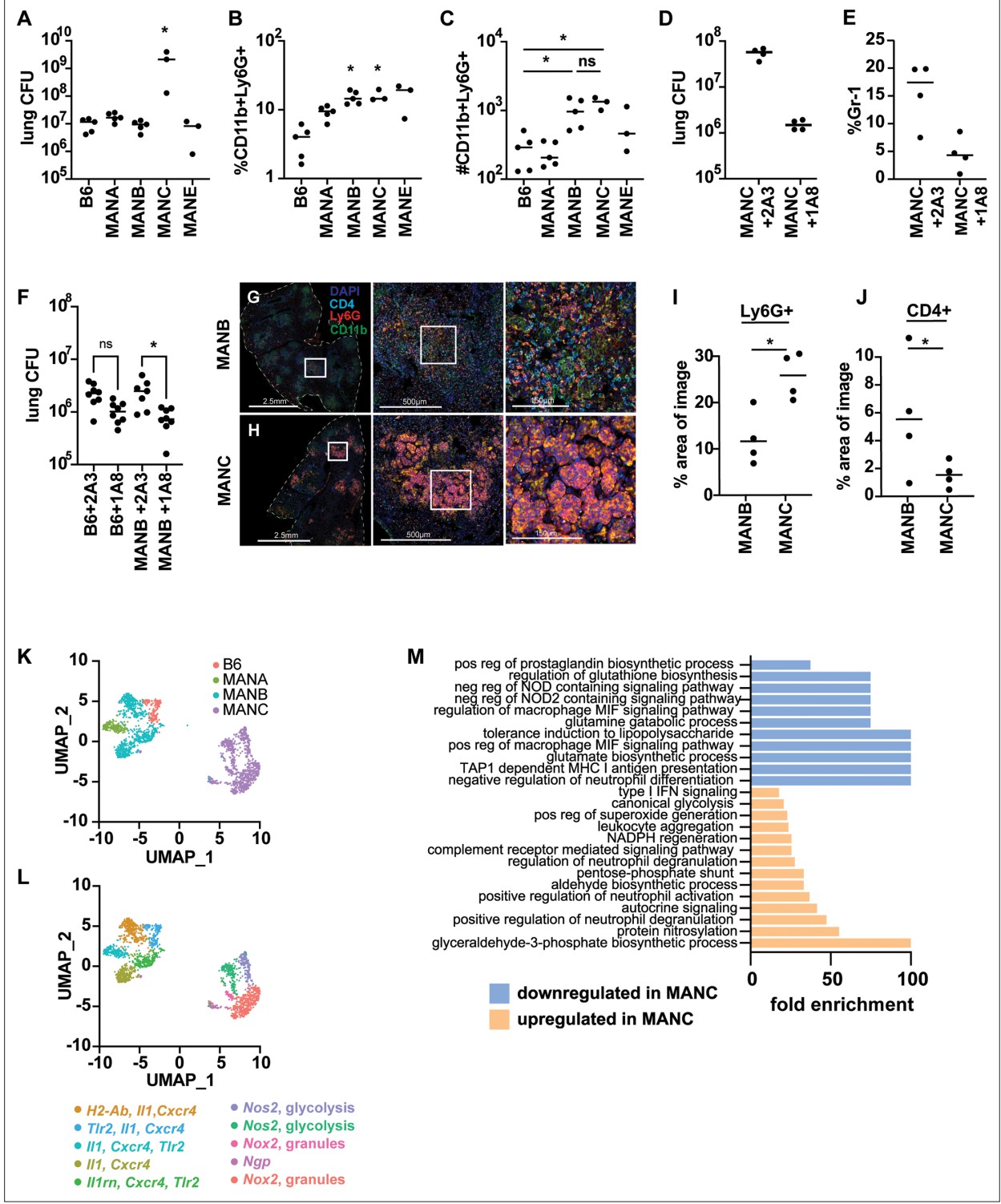

**Figure 4.** Diverse populations of neutrophils are present during infection of Manaus mice. (**A**) Bacterial burden in the lung was assessed in Manaus mice at 25 days post-infection. (**B, C**) Percentages and absolute counts of CD11b+Ly6G+ cells in the lungs at 25 days post-infection were enumerated by flow cytometry. Percentage is relative to the total number of live cells. (**D, E**) MANC and MANB (**F**) mice were treated with neutrophil-depleting antibody (1A8) or isotype control (2A3) every other day starting on day 11 post *Mtb* infection. Bacterial burden and neutrophil numbers were quantified on day 25 post-infection. Immunofluorescent microscopy on fixed lung tissues from MANB (**G**) and MANC (**H**) mice at 21 days post-infection. Area quantification of Ly6G+ (**I**) or CD4+ (**J**) staining in lung sections. (**K, L**) UMAP clustering of neutrophils from scRNAseq data of live cells in the lung at 21 days post *Mtb* infection. Signature functional genes that are characteristic of each cluster are provided with color coding according to cluster. (**M**) Gene

*Figure 4 continued on next page*

*Figure 4 continued*
ontology enrichment analysis of pathways down- or upregulated in MANC neutrophils compared with other neutrophil genotypes shows enrichment of genes involved in neutrophil activation and metabolism. CFU and antibody depletion data are representative of three independent (MANC) or two independent (MANB) experiments. The p values were determined using Mann–Whitney *U* test (**A, I, J**) and Kruskal–Wallis ANOVA (**B, C**), *p < 0.05. scRNAseq data represent a single experiment with pooled tissues from four mice.

The online version of this article includes the following figure supplement(s) for figure 4:

**Figure supplement 1.** Immune cell composition in the lungs of infected mice.

**Figure supplement 2.** T cell subsets and gene expression in wild-derived mice.

In order to elucidate the phenotypic differences of the cell populations in the different wild-derived mouse lines from Manaus, we used single-cell RNAseq (scRNAseq) to identify transcriptional changes in immune cells isolated from lungs of *Mtb*-infected B6, MANA, MANB, and MANC mice at 21 days post-infection. Using signature transcription factor, cytokine, and cell surface marker expression, the dataset was analyzed in the Seurat environment and major immune cell types in the lung were defined. Overall, transcriptional profiles indicated that the cell type composition was relatively similar across genotypes. The exception was an increase in cells identified as neutrophils (*Ly6g+*, *S100a8+*) in both MANB and MANC mice relative to the other genotypes, consistent with our flow cytometry data (***Figure 2—figure supplement 3D-G***).

To identify phenotypic heterogeneity within neutrophils, we subset neutrophils from all genotypes, identified variable features and rescaled the data within the subset, performed PCA dimension reduction and clustering, and visualized the results with the UMAP reduction. Neutrophils clustered into 2 major populations (***Figure 4K***) and at least 10 well-defined subpopulations (***Figure 4L***). B6, MANA, and MANB neutrophils clustered together while MANC neutrophils remained quite distinct. MANB neutrophils exhibited some transcriptional heterogeneity with two distinct neutrophil subpopulations distributed across two clusters (***Figure 4K***). B6 and MANA neutrophils were more homogeneous (***Figure 4K***). MANC clustered separately from neutrophils of any of the other genotypes and exhibited the most within-genotype heterogeneity in neutrophil subsets (***Figure 4K, L***). MANB and MANC mice exhibit comparably elevated numbers of neutrophils in their lungs during *Mtb* infection despite a 2-log difference in bacterial burden (***Figure 4A–C***). We next sought to further characterize transcriptional differences in MANC neutrophils relative to other genotypes across all neutrophil subtypes. Aggregated together, neutrophils from MANC mice were enriched for GO biological processes indicative of neutrophil activation, including positive regulation of neutrophil activation, neutrophil degranulation, superoxide generation, and glycolysis (***Figure 4M***). GO processes downregulated in MANC neutrophils relative to B6 included antigen processing, prostaglandin biosynthesis, and NOD signaling (***Figure 4M***). These data suggest that neutrophils from MANC mice are more activated and more likely to be engaging in processes that result in destruction of host tissue. To confirm these findings, and to determine if the gene expression signature identified to MANC mice is common to other mouse strains that have neutrophil-driven susceptibility to *Mtb*, we compared transcriptional signatures of neutrophils from our B6, MANB, and MANC datasets with transcriptional signatures of neutrophils isolated from *Sp140⁻/⁻* mice. These mice have a type I IFN-driven susceptibility to infection that is also characterized by high levels of neutrophils. Depletion of neutrophils from this model rescues *Sp140⁻/⁻* mice (***Kotov et al., 2023***). We used a series of recently developed signatures for neutrophil differentiation state (***Xie et al., 2020***) to generate scores for neutrophils across a number of parameters: neutrophil activation, NADPH oxidase, and secretory vesicles. In all cases, we found that MANC neutrophils scored higher across the parameters than any other genotype tested, including *Sp140⁻/⁻* (***Figure 4—figure supplement 1***).

Neutrophils from MANC mice express higher levels of neutrophil signature genes, including *S100a8/9*, *Ngp*, and *Ly6g* (***Figure 5A***). These neutrophils also express higher levels of genes associated with pathogen clearance and host tissue damage, including inducible nitric oxide synthase (*Nos2*), NADPH oxidase (*Cybb*), and *Mmp9*. MANC neutrophils' glycolytic signature includes higher expression levels of *Ldha* and *Pfkfb3*. Indeed, our clustering of MANC neutrophils clustered into subsets that were in part driven by differential expression of these marker genes (***Figure 4K***). In contrast, neutrophils from MANA, MANB, and B6 mice expressed higher levels of specific immune-related genes relevant to defense against *Mtb* infection including *Tlr2*, *Il1b*, and MHC II isoforms (*H2-Ab1* and *H2-Eb*) (***Figure 5A***). Other genes that were highly differentially expressed between the lines

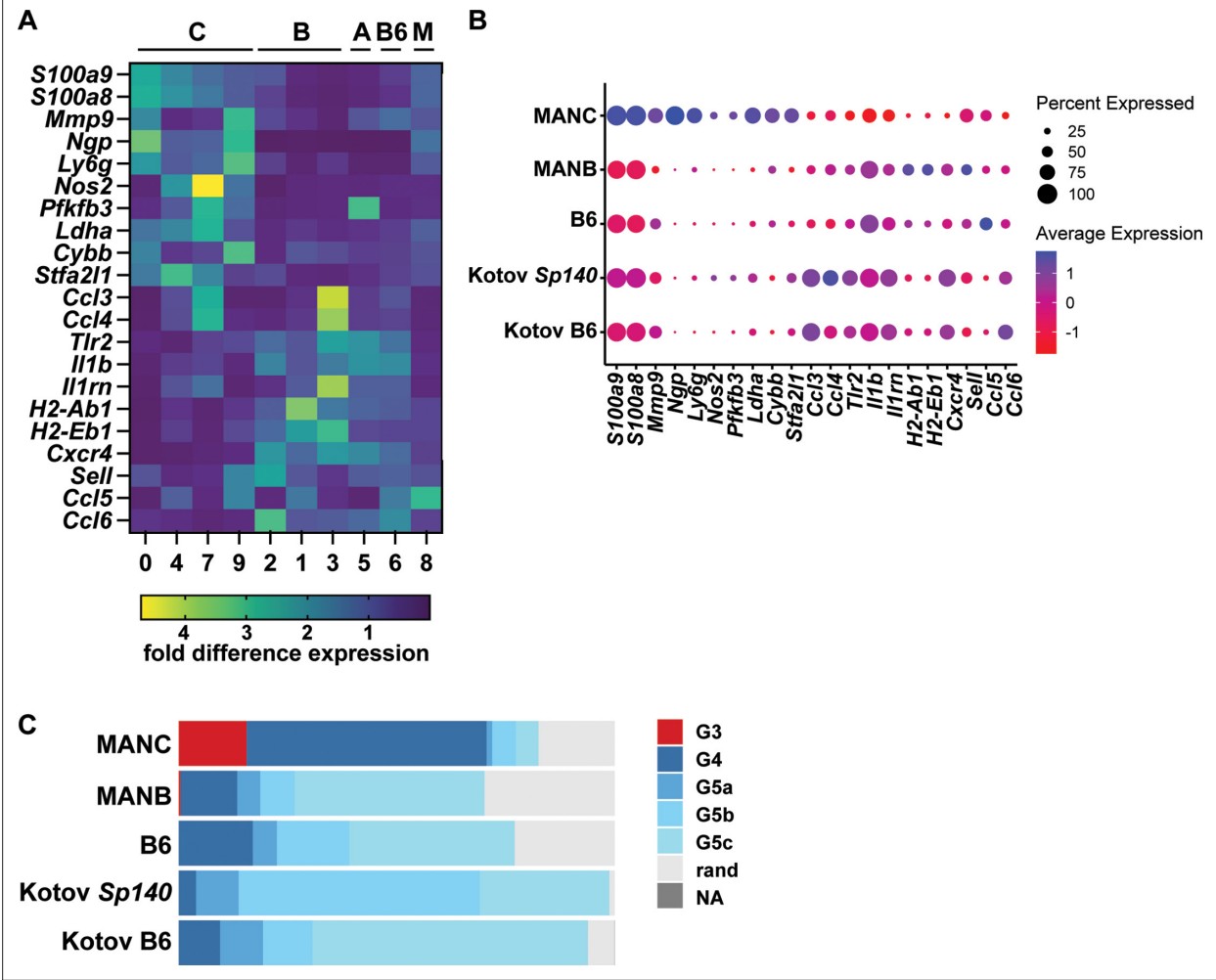

**Figure 5.** Neutrophil signature of MANC mice is unique relative to other susceptible mouse lines. (**A**) Comparison of gene expression between highly expressed genes in the MANC neutrophil clusters with expression in MANB (**B**), MANA (**A**), B6, and clusters of mixed genotype (**M**). (**B**) Comparison of MANC signature genes with expression of genes in the *Sp140⁻/⁻* susceptible mouse line. (**C**) MANC neutrophil signature according to neutrophil developmental stage.

were the chemokines *Ccl3*, *Ccl4*, *Ccl5*, and *Ccl6* (*Figure 5A*). Neutrophils from all lines expressed very low levels of immunosuppressive proteins and cytokines, with the exception of IL-1R antagonist (*Il1rn*), which was highly expressed by a subset of MANB neutrophils (*Figure 5A*). In summary, neutrophils in susceptible MANC mice exhibit a transcriptional program that is unique when compared with other MAN lines or to B6 mice. We compared this signature with the transcriptional signature of *Sp140⁻/⁻* and again found that the MANC transcriptional signature is unique to neutrophils from this line and not a common feature of high bacterial burden or neutrophil-driven pathology during *Mtb* infection (*Figure 5B*).

Premature release of neutrophils at early developmental stages is a hallmark of some inflammatory states, a phenomenon associated with emergency granulopoiesis (*Manz and Boettcher, 2014*). These immature neutrophils, also known as banded neutrophils, have been shown to exhibit a hyperactivated state. Using our scRNAseq data, we evaluated the differentiation state of neutrophils from B6 and Manaus mice and compared them with neutrophils from *Sp140⁻/⁻* mice. Interestingly, we found that neutrophils from the lungs of Mtb-infected MANC mice were uniquely found to exhibit an immature transcriptional phenotype dominated by G3 (immature neutrophils) and G4 (mature neutrophils prior to release from bone marrow). All other genotypes were dominated by mature peripheral neutrophils G5a, G5b, and G5c cells, as expected (*Figure 5C*).

## CD4 T cells from MANC mice are not significantly different from CD4 T cells from other lines

We showed that granuloma-like lesions from MANC mice exhibited a paucity of T cells (*Figure 4G–J*). These data suggest that macrophages in lesions from MANC mice may differentially recruit neutrophils versus T cells, and/or that T cells may have an intrinsic defect in homing to lesions in MANC mice. To identify transcriptional differences in CD4 T cells and macrophage/monocyte populations that distinguish the lines of mice, we turned to our scRNAseq data. After subsetting CD4 T cells from all genotypes, renormalizing gene counts, PCA dimension reduction clustering, and visualizing a UMAP reduction, we identified 13 clusters that correspond to clearly distinct CD4 differentiation states (*Figure 4—figure supplement 2A*). All genotypes possessed naïve CD4 T cells, Th1 differentiated T cells, Ki67+ proliferating T cells, and cytotoxic T cells expressing granzyme B and perforin (cytotoxic CD4, *Figure 4—figure supplement 2B*). B6 mice uniquely had a population of what appear to be terminally differentiated Th1 T cells that express KLRG1 and high levels of IFN-γ (*Figure 4—figure supplement 2B*). These cells have been previously identified as a non-protective T cell subset in B6 mice (*Sakai et al., 2014*; *Cyktor et al., 2013*; *Lindenstrøm et al., 2013*). It is interesting that none of the Manaus lines appear to have this subset of T cells, even in the susceptible MANC line. Other markers of T cell activation and differentiation, including *Cd44*, *Pdcd1*, *Tigit*, and *Cd*69, were not substantially different across mouse lines (*Figure 4—figure supplement 2B*), suggesting that major phenotypic differences in CD4 T cells themselves, separate from their propensity for trafficking into or surviving within the granuloma, do not underlie the susceptibility of MANC mice. We also did not observe major differences in chemokine or chemokine receptor expression of genes known to be required for T cell trafficking into the lung parenchyma or into granulomas.

*CD11b+/Ly6g–* macrophage/monocyte-like cells were also mapped using our scRNAseq data and were found to fall into 14 distinct clusters, many but not all of which were genotype specific (*Figure 6—figure supplement 1A-B*). As expected, alveolar macrophages were observed in every genotype of mouse (*Figure 6—figure supplement 1A, B*). No clear pattern in terms of distribution of cell types emerged from analysis of markers for macrophages (*Adgre1)*, monocytes (*Cx3cr1*), and inflammatory monocytes (*Ly6c*) (*Figure 6A*), although B6 and MANA were dominated by monocytes whereas MANB and MANC mice had clear macrophage populations (*Figure 6—figure supplement 1A, B*). A population of cells with mixed macrophage and monocyte markers was found in every genotype (*Figure 6—figure supplement 1A, B*). Similar to what we previously observed in neutrophils, monocyte/macrophages from MANC mice had a higher propensity to express *Nos2* (*Figure 6A*). No other genes indicative of macrophage function, including *IL1b*, *Arg1*, *Tnf*, and *Cybb*, were differentially expressed across genotypes (*Figure 6A*). Interestingly, MANC macrophages also exhibited very high levels of expression of *S100a9* and *S100a8*, genes more typically associated with neutrophil function (*Figure 6A*).

## Global transcriptional changes in MANC immune cells reveal expanded expression of neutrophil markers across multiple cell lineages

Both macrophages and neutrophils from MANC mice express higher levels of *S100a8* and *S100a9* than cells from other genotypes. To identify differentially expressed genes in MANC mice whose expression patterns are preserved across multiple cell types, we analyzed scRNAseq data. Interestingly, we found that expression of a small number of genes was dramatically increased in MANC mice, and that this signature was dominated by genes that are often thought to be relatively neutrophil specific, including *Ngp* (neutrophil granule protein), *S100a9*, *S100a8*, and *lcn2* (lipocalin 2) (*Figure 6B*). Higher expression of these genes was not solely driven by higher expression levels in neutrophils, as we observed increased expression of these genes across all cell types analyzed, which included major immune cell populations as well as epithelial cells and fibroblasts isolated from lungs (*Figure 6C*). To determine whether expression of highly neutrophil-specific proteins has been observed in other cell types, we examined expression of *Ngp* across cell types in the ImmGen database (*Heng and Painter, 2008*), which holds curated expression data defined by immunological cell type and state. As expected, we found that expression of *Ngp* was observed at highest levels in neutrophils, with dramatically lower levels of expression across other immune cell types (*Figure 6—figure supplement 1C*). Thus, expression of these neutrophil-specific genes across many cell types appears to be a unique feature of MANC mice.

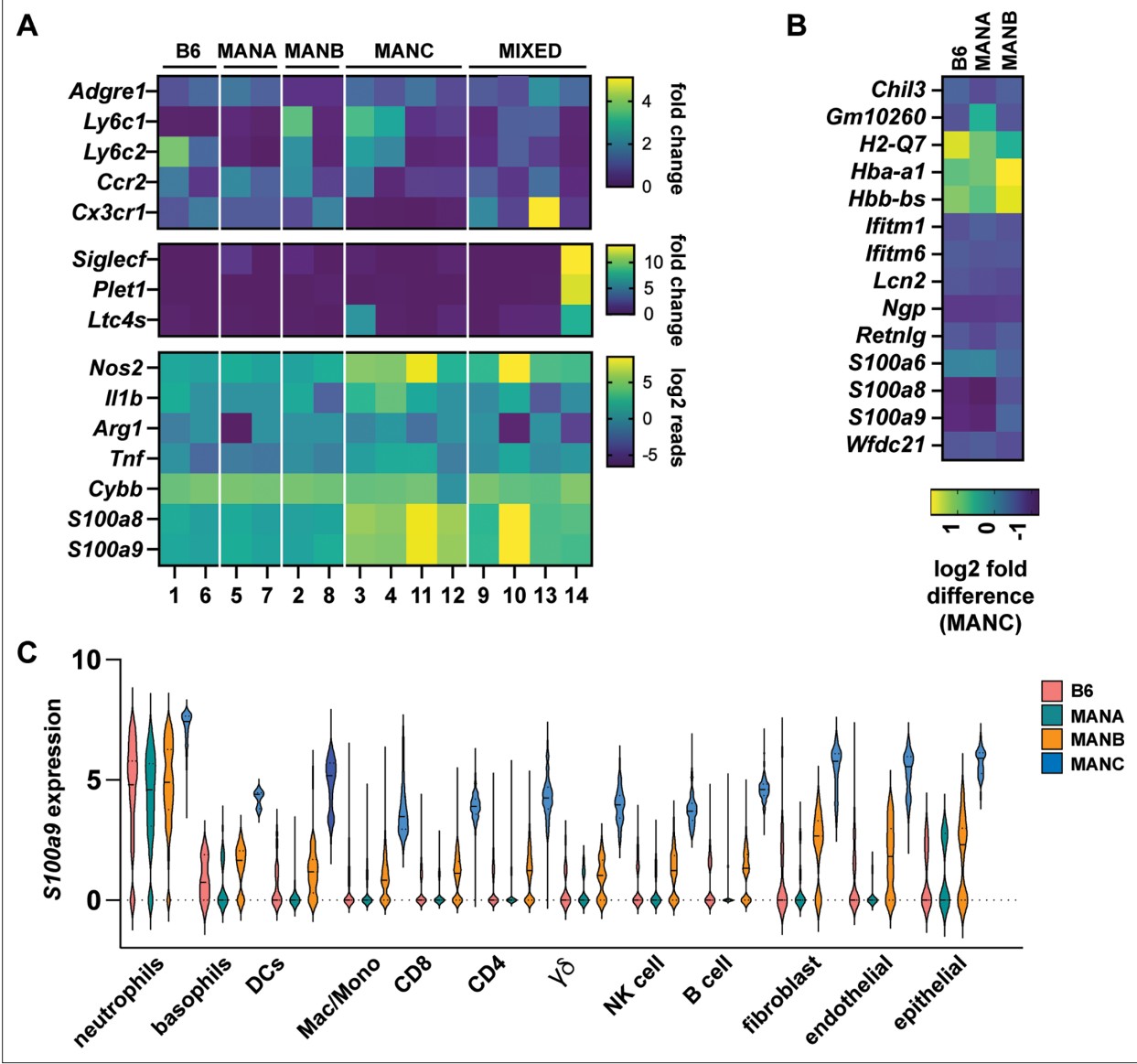

**Figure 6.** Expression of neutrophil markers across multiple cell lineages in MANC mice. (**A**) Heatmap of gene expression of markers of macrophage/ monocyte subsets according to cluster and genotype. (**B**) Expression of indicated genes across all cell types comparing the B6, MANB, and MANC genotypes. (**C**) Violin plot of *S100a9* expression according to cell type and genotype.

The online version of this article includes the following figure supplement(s) for figure 6:

**Figure supplement 1.** Gene expression across immune cells.

## Discussion

Developing immune-based diagnostics and new vaccines for TB has been challenging, likely due in part to the heterogeneity of human responses to disease. Host and bacterial genetics and environmental factors combine to shape the variety of disease outcomes seen in human TB. Furthermore, studies of human immune responses are by their nature rarely able to distinguish mechanisms that are causative of, rather than merely correlative with, disease outcomes. Thus, it has been difficult to answer some of the most important questions in the field. First, we lack an understanding of why some individuals experience a failure of immunity that results in progression to active disease. Further, reliable biomarkers of successful control of TB remain elusive, despite the importance of such biomarkers for guiding the development of new and effective vaccines. The strength of animal models comes from the ability to deeply interrogate novel immunological phenotypes and uncover unrecognized

mechanisms of infection control or susceptibility. Furthermore, the animal model can be used to establish causation based upon observations in humans. For example, there has been significant synergy between mouse and human immune studies for the establishment of type I IFN as an important driver of susceptibility to disease (*Berry et al., 2010*; *Stanley et al., 2007*; *Moreira-Teixeira et al., 2020b*; *Moreira-Teixeira et al., 2020a*). Identifying new immunological mechanisms in mice could lead to human studies focused on these mechanisms. Here we used a new collection of inbred wild mice to study the outcome of infection with *Mtb*. Wild mice exhibited a range in susceptibility at an early time point after infection as well as diversity in the immune responses to infection (*Figures 1 and 2*). Our initial survey of these mice has identified several lines with interesting and novel phenotypes worthy of in-depth investigation, including a unique neutrophil signature associated with susceptibility.

Neutrophils are thought to be an important driver of susceptibility to the outcome of TB in both mice and humans (*Mishra et al., 2017*; *Berry et al., 2010*; *Nandi and Behar, 2011*). It has been proposed that the neutrophils are detrimental to infection due to generation of lung inflammation and NETosis (*Moreira-Teixeira et al., 2020a*). It is still unclear what drives excess accumulation of neutrophils under different conditions. Further, neutrophils can adopt numerous phenotypes, and it is unclear whether all neutrophils are equally destructive during infection or whether specific neutrophil signatures are predictive of disease. Studying the role of neutrophils during *Mtb* infection has been challenging, due to the short lifespan and the subsequent difficulties of culturing neutrophils ex vivo. In addition, there are relatively few models of susceptibility to infection available for analysis. We found that neutrophils are highly abundant in the lungs of infected MANC mice, and that neutrophils drive the susceptibility of this line (*Figure 4A–E*). This finding accords with what has been shown by previous studies of other highly susceptible mice (*Kimmey et al., 2015*; *Mishra et al., 2017*; *Nair et al., 2018*; *Dorhoi et al., 2010*). However, MANB mice also harbor elevated levels of neutrophils, yet control infection as well as the standard B6 control. Histological data from the lungs of MANC mice show Ly6G staining for neutrophils reveals large aggregates of staining within the lesions of MANC mice, whereas neutrophils in MANB mice are more discrete and dispersed throughout the tissue (*Figure 4*). scRNAseq expression analysis shows that neutrophil clusters from MANC mice have a high expression of genes that contribute to neutrophil degranulation as well as superoxide and NO production, indicating potential for tissue destruction during infection. In addition, neutrophils from these mice exhibit a more activated glycolytic transcriptional signature when compared with B6 or other MAN lines. The transcriptional phenotype of protected MANB is strikingly different from MANC, with an expression profile that is associated with protective genes, including *Tlr2* and *Il1b*. We hypothesize that expression of these protective factors combined with a less activated phenotype means that MANB neutrophils may not drive pathology during Mtb infection. Whether these divergent phenotypes reflect differences in inherent immunological responses or are driven by underlying differences in the capacity of these mice to control infection will be an important topic for future study.

We found that neutrophils from highly susceptible MANC mice expressed high levels of *S100a8* and *S100a9* relative to other mouse lines, including the highly susceptible *Sp140*$^{-/-}$ line. *S100a8/a9* expression has been associated with numerous inflammatory states, including severe COVID-19 (*Udeh et al., 2021*; *Silvin et al., 2020*; *Boucher et al., 2024*). In addition, depletion of *S100a8* and *S100a9* was shown to improve chronic control of *Mtb* infection in mice (*Scott et al., 2020*). Finally, expression of these genes has been associated with progression of humans to active disease (*Scott et al., 2020*). Thus, MANC mice may represent an attractive model for studying the role of S100A8/A9 in TB as well as other inflammatory diseases. In addition to neutrophils, we observed increased expression of *S100a8* and *S100a9* (calprotectin) in macrophages and monocytes from MANC mice. Although typically used as a marker for neutrophils, due to high expression of these genes in this cell type, *S100a8* and *S100a9* are known to be expressed by other cell types including epithelial cells. Although it is technically possible that macrophages and monocytes in our dataset appear to express these markers because of phagocytosis of neutrophils, this is unlikely to explain high-level expression of these genes in other cell types.

Another intriguing aspect of the immune response to *Mtb* is the role of IFN-γ during infection. IFN-γ is mainly produced by CD4 T cells early during infection, and a lack of IFN-γ results in extreme susceptibility, with mice succumbing to infection (*Green et al., 2013*; *Flynn et al., 1993*). Furthermore, IFN-γ is required for prevention of bacterial dissemination to other tissues (*Sakai et al., 2016*). However, increased IFN-γ responses in the lung do not seem to correlate with resistance to infection

or improved responses to vaccination and can lead to exacerbated disease (*Tameris et al., 2013*; *Sakai et al., 2016*; *Leal et al., 2001*; *Majlessi et al., 2006*; *Mittrücker et al., 2007*). While local IFN-γ production might be necessary for full control of infection in B6 mice, additional research has shown effects of T cell-mediated IFN-γ independent mechanisms that contribute to control of infection, including through GM-CSF production by CD4 T cells (*Rothchild et al., 2014*; *Rothchild et al., 2017*; *Van Dis et al., 2022*) and the production of IL-17 (*Gallegos et al., 2011*). Several mouse lines in the cohort showed restriction of bacterial growth in the first weeks post-infection equally well but with widely differing IFN-γ+ T cell profiles. SARA mice exhibited enhanced control of bacterial burden (*Figure 1D*), but very low numbers of IFN-γ+ T cells compared to B6 mice (*Figure 2K*, *Figure 2— figure supplement 3*). This suggests either optimal localization of the few IFN-γ producing T cells to the bacterial-containing lesions, thereby allowing for highly efficient induction of local IFN-γ driven protective responses in the lesions, or the presence of IFN-γ or T cell-independent control of infection. Further research is needed to determine the spatial interactions between T cells and other immune cells present in the lesions, including bacilli-containing macrophages. It has previously been shown that homing of *Mtb*-specific T cells to the lung parenchyma is essential for interaction with activated macrophages to induce control of infection; however, signals that allow for T cell penetration into granulomas have not been fully elicited (*Sakai et al., 2016*). Finally, longitudinal studies are needed to determine whether control of infection in SARA mice persists over longer time periods.

The results presented here should inspire more in-depth mechanistic analysis of phenotypes observed in wild mouse lines by our group as well as others in the field. Ultimately, it will be important to identify genetic differences that underlie the differences in capacity to control disease. Inter-crossing of these closely related lines may elucidate genetic loci that are responsible for underlying phenotypes, possibly due to the presence of genetic variants that are absent in standard mouse lines. This strategy has been used with success by the breeding of susceptible C3H and resistant B6 mice to identify the super susceptibility locus 1 (*Sst1*) (*Kramnik et al., 2000*) as well as through the study of genetically diverse cohorts including the CC and DO mice which resulted in the identification of susceptibility markers (*Niazi et al., 2015*; *Smith et al., 2019*).

In addition to the abovementioned phenotypes, many other interesting correlates of immunity were identified in these mice that are worthy of future investigation. For example, we found a correlation between higher B cell numbers and improved resistance to infection; whether B cells may play a role in infection control is a question that can be examined using these lines. These and other phenotypes may be important not only for TB, but for other studies of host immunity. We suggest that these wild mice will be a rich source of phenotype discovery for numerous fields, adding to the heterogeneity of models available for the study of disease.

## Materials and methods

### Mice

Wild-derived mouse lines from the Manaus (MANA, MANB, MANC, MANE), Saratoga Springs (SARA, SARB, SARC), Edmonton (EDMA, EDMC, EDMD), Tucson (TUCA, TUCB, TACC), and Gainesville (GAIA, GAIB, GAIC) locations were obtained from the Nachman laboratory (UC, Berkeley). Several lines from these cohorts have since been deposited to The Jackson Laboratory (Bar Harbor, ME) as a public resource (ManA/NachJ #035354; ManB/NachJ #035355; ManE/NachJ #035365; SarA/ NachJ #035346; SarB/NachJ #035347; SarC/NachJ #035348; TUCA/NachJ #035358; TUCB/NachJ #035359; GaiC/NachJ #035352). C57BL/6J (#000664) mice were obtained from The Jackson Laboratory and bred in-house. For mouse weight measurements, male mice were housed singly in a cage to avoid social dominance effects on weights. Female mice were group housed and body weights were measured at 7 weeks old and prior to infection with *Mtb*.

### Bacterial culture

Low passage stocks of *Mtb* Erdman were grown in Middlebrook 7H9 liquid medium supplemented with 10% albumin-dextrose-saline or on solid 7H10 agar plates supplemented with 10% Middlebrook oleic acid, albumin, dextrose, and catalase (BD Biosciences) and 0.4% glycerol. Frozen stocks of *Mtb* were made from liquid cultures and used for all in vivo infection experiments.

## In vivo infections

Mice were infected via the aerosol route with *Mtb* strain Erdman. Aerosol infection was done using a nebulizer and full-body inhalation exposure system (Glas-Col, Terre Haute, IN). Bacterial culture diluted in sterile water was loaded into the nebulizer calibrated to deliver ~400 bacteria per mouse, as measured by CFU in the lungs 1 day following infection (data not shown). Neutrophil depletion experiments were performed as described in *Nandi and Behar, 2011* using the anti-Ly6G monoclonal Ab (clone 1A8; BioXCell) or isotype control (clone 2A3, BioXCell). Two hundred micrograms of Ab were injected every other day starting at day 11 post-infection. Mice were sacrificed at 21 or 25 days post-infection to measure CFU and immune responses in the lungs. For bacterial enumeration, the largest lung lobe was homogenized in PBS supplemented with 0.05% Tween 80, and serial dilutions were plated on 7H10 plates. CFU were counted 21 days after plating. The three smallest lung lobes were harvested into complete RPMI (supplemented with 10% fetal bovine serum, 2 mM L-glutamine, 100 U/ml penicillin/streptomycin, 1 mM sodium pyruvate, 20 mM HEPES, 55 µM 2-mercaptoethanol), and strained through a 40-µm Falcon strainer. Cells were washed, stained with antibodies used for cytokine and surface marker staining: live/dead (L34970; Thermo Fisher Scientific), CD4, CD3, CD8a, CD11b, B220, Ly-6G (100509, 100219, 100707, 101237, 103233, 127627, respectively; BioLegend), and fixed/permeabilized with BD Cytofix/Cytoperm Fixation/Permeabilization Solution Kit (554714; Thermo Fisher Scientific) before staining with antibody specific for IFN-γ (505816; BioLegend). Data were collected using a BD LSR Fortessa flow cytometer and analyzed using FlowJo Software (Tree Star, Ashland, OR). All cell populations were gated on live cells. CD45 antibodies did not provide consistent staining for all wild mice genotypes. CD4 and CD8 cell populations were identified from CD3+ cells, followed by IFN-γ staining. Neutrophils were identified as CD11b+Ly6G+ and macrophages/monocytes as CD11b+Ly6G− (*Figure 2—figure supplement 2*).

## IL-1 bioactivity reporter assay

The second largest lung lobe from infected mice was harvested into complete RPMI supplemented with Pierce proteinase inhibitor EDTA-free (Thermo Fisher Scientific), dissociated, and strained through a 40-µM Falcon strainer. Cells and debris were removed by a low-speed centrifugation (300 × *g*) followed by a high-speed centrifugation (2000 × *g*) and stored at −80°C. Samples were assayed using HEK-Blue IL-1R cells (InvivoGen, San Diego, CA). A total of $3.75 \times 10^4$ cells per well were plated in 96-well plates and allowed to adhere overnight in DMEM supplemented with 10% FBS, 2 mM L-glutamine. Reporter cells were treated overnight with 100 µl of sample (consisting of 50 µl of cell-free lung homogenates and 50 µl of media), or recombinant mouse IL-1β (R&D Systems 401-ML-005). Assays were developed using QUANTI-Blue (InvivoGen) according to the manufacturer's protocols and measured using a SpectraMax M2 microplate reader (Molecular Devices, San Jose, CA).

## IFN-β bioactivity reporter assay

Interferon responsive ISRE-L929 cells (gift from D. Portnoy, UC Berkeley) were cultured in ISRE media (DMEM, 2 mM glutamine, 1 mM pyruvate, 10% heat inactivated FBS, and penicillin–streptomycin). The presence of type I interferon was assessed using lung homogenate as described above. Homogenates were applied in various dilutions to the interferon-responsive ISRE-L929 cells ($5 \times 10^4$ cells/well) in white, 96-well, tissue culture-treated plates (Thermo Scientific Nunc). Cells were incubated for 4 hr, media aspirated, and cells were lysed with 30 µl Luciferase cell culture lysis reagent (Promega, Madison, WI). Finally, 100 µl of luciferase substrate solution (Promega) was added to each well and luminescence was measured using a Spectramax L luminometer (Molecular Devices).

## QRT-PCR

The second largest lung lobe from infected mice was harvested into 500 µl RNAlater solution (Invitrogen Life Technologies, Carlsbad, CA) and stored at −80°C for qRT-PCR analysis. Tissues were transferred to 500 µl TRIzol (Invitrogen Life Technologies) and homogenized. Total RNA was extracted using chloroform, and the aqueous layer was further purified using RNeasy spin columns (QIAGEN, Hilden, Germany). For qPCR, cDNA was generated from 1 mg RNA using Superscript III (Invitrogen Life Technologies) and oligo(dT) primers. Select genes were analyzed using Power SYBR Green qPCR master mix (Thermo Scientific). Each sample was analyzed in triplicate on a CFX96 real-time PCR

detection system (Bio-Rad, Hercules, CA). CQ values were normalized to values obtained for GAPDH, and relative changes in gene expression were calculated as $\Delta\Delta C_Q$.

## In vitro infections

Bone marrow was obtained from B6 and wild-derived mice and cultured in DMEM with 10% FBS, 2 mM L-glutamine, and 10% supernatant from 3T3-M-CSF cells for 6 days with feeding on day 3 to generate BMDM. BMDM were plated into 96- or 24-well plates with $5 \times 10^4$ and $3 \times 10^5$ macrophages/well, respectively, and were allowed to adhere and rest for 24 hr. BMDM were then treated with vehicle or recombinant mouse IFN-γ at 6.25 ng/ml (485-MI, R&D Systems) overnight and infected with *Mtb* Erdman strain at a multiplicity of infection of 5 in DMEM supplemented with 5% horse serum and 5% FBS, unless otherwise noted. After a 4-hr phagocytosis period, media was replaced with DMEM supplemented with 10% FBS, 2 mM glutamine, and 10% M-CSF. For IFN-γ-pretreated wells, IFN-γ was added post-infection at the same concentration. For enumeration of CFU, infected BMDM were washed with PBS and lysed in water containing 0.1% Triton X-100 for 10 min. Serial dilutions were prepared in PBS with 0.05% Tween 80 and were plated onto 7H10 plates. Supernatant nitrite was measured by Griess assay as a proxy for NO production by mixing supernatant 1:1 with a solution of 0.1% napthylethylenediamine, 1% sulfanilamide, and 2% phosphoric acid and measuring absorbance at 546 nm with a SpectraMax M2 microplate reader (Molecular Devices). Not every mouse line is included in in vitro profiling due to small litters and lack of availability of mice.

## Immunofluorescent microscopy

Lung tissues were fixed in 10% neutral-buffered formalin at room temperature for 24 hr. Tissue embedding and sectioning at 5 μM was performed by the Center for Genomic Pathology Laboratory (UC Davis) and staining for anti-CD4 (D7D2Z, Cell Signaling Technology, Danvers, MA), anti-CD11b (M1/70, BioLegend), and anti-Ly6G (1A8, BioLegend) were performed by NanoString Technologies (Seattle, WA). Analysis of images for quantification was performed using ImageJ. Images were quantified at ×100 magnification.

## scRNAseq and transcriptional analysis

The second largest lung lobes were pooled from five infected mice, tissue dissociated in RPMI 1640 containing liberase and DNase I in gentleMACS C tubes using the gentleMACS dissociator, strained through a 70-μM Falcon cell strainer in RPMI 1640 and counted. A total of 10,000 cells per sample were used for scRNAseq according to the 10x Genomics protocol. Briefly, single-cell suspensions were partitioned into Gel Beads in emulsion using the Chromium Next GEM Single Cell v3.1 system. Barcoded primers were used to reverse-transcribe poly-adenylated mRNA to produce barcoded, full-length cDNA. Purified DNA was amplified, fragmented, and amplified again to attach Illumina adapter sequences by the Functional Genomics Laboratory at UC, Berkeley. Libraries were sequenced on an Illumina NovaSeq S1 demultiplexed by the Vincent J. Coates Genomics Sequencing Laboratory at UC, Berkeley. Reads were aligned to the mouse transcriptome mm10 with the 10x Cell Ranger analysis pipeline using the Savio computational cluster at UC, Berkeley. Data can be found under GSE242343. After filtering, barcode counting, and unique molecular identifier counting, the Seurat 4.0.6 toolkit (Satija lab; *Hao et al., 2021*) was used to preprocess the data. Dimension reduction (PCA) and κ-nearest neighbor graph construction (25 PCs) was followed by clustering (resolution 0.75) and visualization of the data with UMAP reduction. The clustered data were manually assigned a cell type using signature genes and gene expression and visualization was explored using Seurat (*Supplementary file 1*). GO terms were analyzed using the top 200 differentially expressed genes of each dataset. For neutrophil and macrophage/monocyte analysis, the relevant cells were subset, variable features were re-identified, and the data rescaled within the subset. Dimension reduction (PCA) and κ-nearest neighbor graph construction (first 25 PCs), clustering (resolution 0.75) was followed by visualization and data exploration with the UMAP reduction. Differential gene expression was performed with Seurat's FindMarkers function. Gene ontology analysis was performed with topGO (v 2.48.0). The baseline of expression of neutrophil-related genes in mouse immune cell types (*Figure 6—figure supplement 1*) was obtained from the Immunological Genome Project MyGeneset application (http://rstats.immgen.org/MyGeneSet_New/index.html) using the ImmGen ULI RNA-seq dataset.

## Neutrophil staging

Neutrophils were staged using single-cell transcriptional signatures developed by *Xie et al., 2020* (GSE137539) and a classifier constructed following methods developed by *Kim et al., 2022* using singleCellNet v. 0101 (*Tan and Cahan, 2019*). The classifier was trained with the scn_train() function and *nTopGenes* = 100, *nTopGenePairs* = 50, *nRand* = 50, *nTrees* = 1000. For comparison with wild mice neutrophils, B6 and Sp140$^{-/-}$ neutrophils were integrated from *Kotov et al., 2023* (GSE216023). Neutrophils from our dataset and Kotov's were subset into one unique Seurat object for each genotype and dataset. Integration anchors were found with *FindIntegrationAnchors()* followed by *IntegrateData()*. The combined, integrated dataset was then scaled, normalized, clustered, and inspected following dimensionality reduction with PCA and UMAP. The singleCellNet functions *sc_predict()* and *get_cate()* were used to classify cells in our dataset and to assign a neutrophil stage to each cell. Code for construction of Xie's neutrophil staging classifier, integration of the neutrophil datasets, and the assignment of neutrophil stages is available at GitHub (https://github.com/dmfox244/wm_neut_staging copy archived at *Fox, 2024*).

## Statistical analysis

Data are presented as mean values, and error bars represent SD. Symbols represent individual animals. The statistical tests used are denoted in the legend of the corresponding figure for each experiment. Analysis of statistical significance was performed using GraphPad Prism 8 (GraphPad, La Jolla, CA), and p <0.05 was considered significant.

# Acknowledgements

We want to thank the members of the Stanley, Cox, and Vance labs for helpful discussions. We thank Justin Y Choi at the Functional Genomics Laboratory (UC Berkeley) for scRNAseq library preparation and sequencing, QB3 Genomics and the Vincent J Coates Genomics Sequencing Laboratory (UC Berkeley) for scRNAseq sample sequencing, Qian Chen at the Center for Genomic Pathology Laboratory (UC Davis) for tissue embedding and sectioning, and Wei Yang at NanoString Technologies for immunofluorescent microscopy staining. This work was supported by NIH R21AI146810 and NIH P01 AI063302 to SAS and by NIH RO1 GM074245 and NIH R01 GM127468 to MWN as well as a NSF Graduate Research Fellowship to MMR-C.

# Additional information

### Competing interests

Russell E Vance: R.E.V. is on the scientific advisory boards of Tempest Therapeutics and X-biotix. Sarah A Stanley: Sarah Stanley is on the scientific advisory board of X-biotix Therapeutics. The other authors declare that no competing interests exist.

### Funding

| Funder | Grant reference number | Author |
|---|---|---|
| National Institute of Allergy and Infectious Diseases | R21AI146810 | Sarah A Stanley |
| National Institute of Allergy and Infectious Diseases | P01 AI063302 | Sarah A Stanley |
| National Institute of General Medical Sciences | GM127468 | Michael W Nachman |
| National Institute of General Medical Sciences | R01 GM074245 | Michael W Nachman |
| National Science Foundation | GRFP | Marietta M Ravesloot-Chavez |

| Funder | Grant reference number | Author |
|--------|------------------------|--------|

The funders had no role in study design, data collection, and interpretation, or the decision to submit the work for publication.

## Author contributions

Marietta M Ravesloot-Chavez, Conceptualization, Data curation, Formal analysis, Validation, Investigation, Visualization, Methodology; Erik Van Dis, Andrea Anaya-Sanchez, Investigation, Writing - review and editing; Douglas Fox, Data curation, Formal analysis, Investigation, Writing - review and editing; Scott Espich, Xammy Huu Nguyenla, Sagar Rawal, Helia Samani, Investigation; Mallory Ballinger, Henry F Thomas, Resources; Dmitri I Kotov, Formal analysis; Russell E Vance, Formal analysis, Supervision; Michael W Nachman, Conceptualization, Resources, Supervision, Writing - review and editing; Sarah A Stanley, Conceptualization, Data curation, Formal analysis, Supervision, Funding acquisition, Methodology, Writing - original draft, Project administration, Writing - review and editing

## Author ORCIDs

Marietta M Ravesloot-Chavez ⓘ http://orcid.org/0000-0001-5129-7156
Xammy Huu Nguyenla ⓘ https://orcid.org/0000-0002-7532-8356
Dmitri I Kotov ⓘ https://orcid.org/0000-0001-7843-1503
Russell E Vance ⓘ https://orcid.org/0000-0002-6686-3912
Sarah A Stanley ⓘ https://orcid.org/0000-0002-4182-9048

## Ethics

All procedures involving the use of mice were approved by the University of California (UC), Berkeley, Institutional Animal Care and Use Committee (AUP-2015-09-7979-2). All protocols conform to federal regulations and the National Research Council's Guide for the Care and Use of Laboratory Animals.

Reviewer #1 (Public review): https://doi.org/10.7554/eLife.102441.2.sa1
Reviewer #2 (Public review): https://doi.org/10.7554/eLife.102441.2.sa2
Author response https://doi.org/10.7554/eLife.102441.2.sa3

# Additional files

## Supplementary files

Supplementary file 1. Cellular expression markers used for scRNAseq cluster assignment following unbiased clustering.

MDAR checklist

## Data availability

ScRNAseq data has been deposited in GEO, GSE242343.

The following dataset was generated:

| Author(s) | Year | Dataset title | Dataset URL | Database and Identifier |
|-----------|------|---------------|-------------|-------------------------|
| Stanley AS, Ravesloot-Chavez MM | 2025 | Gene expression profile at single cell level on cells from the lungs of mice infected with Mycobacterium tuberculosis | https://www.ncbi.nlm.nih.gov/geo/query/acc.cgi?acc=GSE242343 | NCBI Gene Expression Omnibus, GSE242343 |

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
