## [Editor Report · eLife Assessment]

This study provides **valuable** insights into the host's variable susceptibility to *Mycobacterium tuberculosis*, using a novel collection of wild-derived inbred mouse lines from diverse geographic locations, along with immunological and single-cell transcriptomic analyses. While the data are **convincing**, a deeper mechanistic investigation into neutrophil subset functions would have further enhanced the study. This work will interest microbiologists and immunologists in the tuberculosis field.

---

## [Referee Report · Reviewer #1 (Public review)]

Summary:

This study investigated the heterogeneous responses to *Mycobacterium tuberculosis* (Mtb) in 19 wild-derived inbred mouse strains collected from various geographic locations. The goal of this study is to identify novel mechanisms that regulate host susceptibility to Mtb infection. Using the genetically resistant C57BL/6 mouse strain as the control, they successfully identified a few mouse strains that revealed higher bacterial burdens in the lung, implicating increased susceptibility in those mouse strains. Furthermore, using flow cytometry analysis, they discovered strong correlations between CFU and various immune cell types, including T cells and B cells. The higher neutrophil numbers correlated with significantly higher CFU in some of the newly identified susceptible mouse strains. Interestingly, MANB and MANC mice exhibited comparable numbers of neutrophils but showed drastically different bacterial burdens. The authors then focused on the neutrophil heterogeneity and utilized a single-cell RNA-seq approach, which led to identifying distinct neutrophil subsets in various mouse strains, including C57BL/6, MANA, MANB, and MANC. Pathway analysis on neutrophils in susceptible MANC strain revealed a highly activated and glycolytic phenotype, implicating a possible mechanism that may contribute to the susceptible phenotype. Lastly, the authors found that a small group of neutrophil-specific genes are expressed across many other cell types in the MANC strain.

Strengths:

This manuscript has many strengths.

(1) Utilizing and characterizing novel mouse strains that complement the current widely used mouse models in the field of TB. Many of those mouse strains will be novel tools for studying host responses to Mtb infection.

(2) The study revealed very unique biology of neutrophils during Mtb infection. It has been well-established that high numbers of neutrophils correlate with high bacterial burden in mice. However, this work uncovered that some mouse strains could be resistant to infection even with high numbers of neutrophils in the lung, indicating the diverse functions of neutrophils. This information is important.

Weaknesses:

The weaknesses of the manuscript are that the work is relatively descriptive. It is unclear whether the neutrophil subsets are indeed functionally different. While single-cell RNA seq did provide some clues at transcription levels, functional and mechanistic investigations are lacking. Similarly, it is unclear how highly activated and glycolytic neutrophils in MANC strain contribute to its susceptibility.

---

## [Referee Report · Reviewer #2 (Public review)]

Summary:

These studies investigate the phenotypic variability and roles of neutrophils in tuberculosis (TB) susceptibility by using a diverse collection of wild-derived inbred mouse lines. The authors aimed to identify new phenotypes during *Mycobacterium tuberculosis* infection by developing, infecting, and phenotyping 19 genetically diverse wild-derived inbred mouse lines originating from different geographic regions in North America and South America. The investigators achieved their main goals, which were to show that increasing genetic diversity increases the phenotypic spectrum observed in response to aerosolized *M. tuberculosis*, and further to provide insights into immune and/or inflammatory correlates of pulmonary TB. Briefly, investigators infected wild-derived mice with aerosolized *M. tuberculosis* and assessed early infection control at 21 days post-infection. The time point was specifically selected to correspond to the period after infection when acquired immunity and antigen-specific responses manifest strongly, and also early susceptibility (morbidity and mortality) due to *M. tuberculosis* infection has been observed in other highly susceptible wild-derived mouse strains, some Collaborative Cross inbred strains, and approximately 30% of individuals in the Diversity Outbred mouse population. Here, the investigators normalized bacterial burden across mice based on inoculum dose and determined the percent of immune cells using flow cytometry, primarily focused on macrophages, neutrophils, CD4 T cells, CD8 T cells, and B cells in the lungs. They also used single-cell RNA sequencing to identify neutrophil subpopulations and immune phenotypes, elegantly supplemented with in vitro macrophage infections and antibody depletion assays to confirm immune cell contributions to susceptibility. The main results from this study confirm that mouse strains show considerable variability to *M. tuberculosis* susceptibility. Authors observed that enhanced infection control correlated with higher percentages of CD4 and CD8 T cells, and B cells, but not necessarily with the percentage of interferon-gamma (IFN-γ) producing cells. High levels of neutrophils and immature neutrophils (band cells) were associated with increased susceptibility, and the mouse strain with the most neutrophils, the MANC line, exhibited a transcriptional signature indicative of a highly activated state, and containing potentially tissue-destructive, mediators that could contribute to the strain's increased susceptibility and be leveraged to understand how neutrophils drive lung tissue damage, cavitation, and granuloma necrosis in pulmonary TB.

Strengths:

The strengths are addressing a critically important consideration in the tuberculosis field - mouse model(s) of the human disease, and taking advantage of the novel phenotypes observed to determine potential mechanisms. Notable strengths include,

(1) Innovative generation and use of mouse models: Developing wild-derived inbred mice from diverse geographic locations is innovative, and this approach expands the range of phenotypic responses observed during *M. tuberculosis* infection. Additionally, the authors have deposited strains at The Jackson Laboratory making these valuable resources available to the scientific community.

(2) Potential for translational research: The findings have implications for human pulmonary TB, particularly the discovery of neutrophil-associated susceptibility in primary infection and/or neutrophil-mediated disease progression that could both inform the development of therapeutic targets and also be used to test the effectiveness of such therapies.

(3) Comprehensive experimental design: The investigators use many complementary approaches including in vivo *M. tuberculosis* infection, in vitro macrophage studies, neutrophil depletion experiments, flow cytometry, and a number of data mining, machine learning, and imaging to produce robust and comprehensive analyses of the wild-derives d strains and neutrophil subpopulations in 3 weeks after *M. tuberculosis* infection.

Weaknesses:

The manuscript and studies have considerable strengths and very few weaknesses. One minor consideration is that phenotyping is limited to a single limited-time point; however, this time point was carefully selected and has a strong biological rationale provided by investigators. This potential weakness does not diminish the overall findings, exciting results, or conclusions.

---

## [Author Response]

**Reviewer #1 (Public review):**
[…] Strengths:This manuscript has many strengths.(1) Utilizing and characterizing novel mouse strains that complement the current widely used mouse models in the field of TB. Many of those mouse strains will be novel tools for studying host responses to Mtb infection.(2) The study revealed very unique biology of neutrophils during Mtb infection. It has been well-established that high numbers of neutrophils correlate with high bacterial burden in mice. However, this work uncovered that some mouse strains could be resistant to infection even with high numbers of neutrophils in the lung, indicating the diverse functions of neutrophils. This information is important.

We are grateful for the reviewer’s thoughtful consideration of our work and appreciate their comment that our mouse strains can benefit the models available in the TB field. We further appreciate the recognition of the importance of neutrophil diversity during Mtb infection.

Weaknesses:The weaknesses of the manuscript are that the work is relatively descriptive. It is unclear whether the neutrophil subsets are indeed functionally different. While single-cell RNA seq did provide some clues at transcription levels, functional and mechanistic investigations are lacking.

We appreciate this comment and agree that further research needs to be done on the functionality of the neutrophils to discover mechanistic differences between the mouse genotypes. Out attempts at extracting sufficient RNA from sorted neutrophils from the mouse lungs were unsuccessful. However, future attempts at comparing RNA expression between mouse genotypes as well as proteomic data are necessary to determine the mechanistic differences in neutrophil biology in these mice.

Similarly, it is unclear how highly activated and glycolytic neutrophils in MANC strain contribute to its susceptibility.

This is a fair comment and we agree that it is still unclear how these neutrophils contribute to MANC susceptibility. Growing the neutrophils ex vivo and infecting them with Mtb is technically challenging, due to the slow growth of Mtb and the short lifespan of the neutrophils. As mentioned in the comment above, future in vivo characterization and RNA expression studies will be necessary to address these questions.

**Reviewer #2 (Public review):**
[…] Strengths:The strengths are addressing a critically important consideration in the tuberculosis field - mouse model(s) of the human disease, and taking advantage of the novel phenotypes observed to determine potential mechanisms. Notable strengths include,(1) Innovative generation and use of mouse models: Developing wild-derived inbred mice from diverse geographic locations is innovative, and this approach expands the range of phenotypic responses observed during *M. tuberculosis* infection. Additionally, the authors have deposited strains at The Jackson Laboratory making these valuable resources available to the scientific community.(2) Potential for translational research: The findings have implications for human pulmonary TB, particularly the discovery of neutrophil-associated susceptibility in primary infection and/or neutrophil-mediated disease progression that could both inform the development of therapeutic targets and also be used to test the effectiveness of such therapies.(3) Comprehensive experimental design: The investigators use many complementary approaches including in vivo *M. tuberculosis* infection, in vitro macrophage studies, neutrophil depletion experiments, flow cytometry, and a number of data mining, machine learning, and imaging to produce robust and comprehensive analyses of the wild-derives d strains and neutrophil subpopulations in 3 weeks after *M. tuberculosis* infection.

We thank the reviewer for their thorough and thoughtful assessment of our study. We appreciate the recognition that this mouse model can become a resource and can benefit the study of different immune responses to Mtb infection as well as be informative for studying human TB. We further appreciate their comment that the complementary approaches we have used to characterized the mouse phenotypes strengthens this study.

Weaknesses:The manuscript and studies have considerable strengths and very few weaknesses. One minor consideration is that phenotyping is limited to a single limited-time point; however, this time point was carefully selected and has a strong biological rationale provided by investigators. This potential weakness does not diminish the overall findings, exciting results, or conclusions.

We thank the reviewer for pointing out that a single time point has been studied, and that this time point is biologically relevant. We agree that additional time points, including later time points that address systemic dissemination, should be included in future studies.